# A Fast and Reliable Solution to PnP, Using Polynomial Homogeneity and a Theorem of Hilbert

**DOI:** 10.3390/s23125585

**Published:** 2023-06-14

**Authors:** Daniel Keren, Margarita Osadchy, Amit Shahar

**Affiliations:** Department of Computer Science, University of Haifa, Haifa 3498838, Israel; rita@cs.haifa.ac.il (M.O.); ashaha16@campus.haifa.ac.il (A.S.)

**Keywords:** the PnP problem, polynomial optimization

## Abstract

One of the most-extensively studied problems in three-dimensional Computer Vision is “Perspective-n-Point” (PnP), which concerns estimating the pose of a calibrated camera, given a set of 3D points in the world and their corresponding 2D projections in an image captured by the camera. One solution method that ranks as very accurate and robust proceeds by reducing PnP to the minimization of a fourth-degree polynomial over the three-dimensional sphere $S^{3}$. Despite a great deal of effort, there is no known fast method to obtain this goal. A very common approach is solving a convex relaxation of the problem, using “Sum Of Squares” (SOS) techniques. We offer two contributions in this paper: a faster (by a factor of roughly 10) solution with respect to the state-of-the-art, which relies on the polynomial’s homogeneity; and a fast, guaranteed, easily parallelizable approximation, which makes use of a famous result of Hilbert.

## 1. Introduction

The PnP problem is a classic computer vision problem that involves estimating the pose of a calibrated camera in 3D space, given a set of correspondences between 3D points in the world and their 2D projections in the camera image. The goal is to determine the camera’s position and orientation (i.e., its pose) relative to the 3D points in the world. The PnP problem has many important applications in robotics, augmented reality, and computer vision. There are several algorithms that have been developed to solve the PnP problem, including the classic Direct Linear Transformation (DLT) [1] algorithm and more recent algorithms such as the Efficient Perspective-n-Point (EPnP) [2] algorithm and the Universal Perspective-n-Point (UPnP) [3] algorithm. These algorithms are based on different optimization techniques, and they have been shown to provide better accuracy and robustness than the classic DLT algorithm. Other work includes [2,4,5,6,7,8].

We next present the notations used hereafter and outline the approach we followed, which yielded a polynomial optimization problem. Thus, we continue a long line of research, starting with [4], with some of the more recent work including [5,6,9,10]. As we shall elaborate in the following sections, our contribution is twofold: we offer a method to solve the optimization problem, which is much faster than previous work, as well as a guaranteed approximation, which is also easily parallelizable.

Given 2D projections of 3D points whose real-world coordinates ${\left\{\left(x_{i},y_{i},z_{i}\right)\right\}}_{i=1}^{n}$ are known, one seeks to determine the camera position and angle, i.e., a translation vector *T* and a rotation matrix *R*, relative to the world coordinate frame, which provide an optimal fit between the 3D points and their projections on the camera image plane.

Denoting by $u_{i}$ the unit vectors in the direction of the projections of $p_{i}$, we obtain the following very-well-known expression to minimize [4]:(1)$$\sum_{i}{∥Q_{i}\left(Rp_{i}+T\right)∥}^{2},\;\;Q_{i}≜I-u_{i}u_{i}^{t}$$

Geometrically (see Figure 1, borrowed from [11]), rotating the point set $\left\{p_{i}\right\}$ by the optimal *R* and translating by the optimal *T* maximize the sum of squared norms of their respective projections on the lines spanned by $\left\{u_{i}\right\}$ (i.e., the points are optimally aligned with the respective “lines of sight”).

To minimize Equation (Equation 1), one first differentiates by *T* and equates to 0; this yields *T* as a function of *R*. Substituting back in Equation (Equation 1) yields:(2)$$\mathrm{Minimize}\;\;R^{t}PR,\;\;P=\sum_{i}\left(C_{i}^{t}-A^{t}\right)Q_{i}\left(C_{i}-A\right),\;\;A≜{\left(\sum_{i}Q_{i}\right)}^{-1}\left(\sum_{i}Q_{i}C_{i}\right)$$
where *R* is the “flattened” (9×1) rotation matrix, and
$$C_{i}=\left(\begin{array}{ccccccccc} x_{i} & y_{i} & z_{i} & 0 & 0 & 0 & 0 & 0 & 0\\ 0 & 0 & 0 & x_{i} & y_{i} & z_{i} & 0 & 0 & 0\\ 0 & 0 & 0 & 0 & 0 & 0 & x_{i} & y_{i} & z_{i} \end{array}\right).$$

Therefore, we are now faced with the problem of minimizing a quadratic polynomial in the nine elements of a rotation matrix, which are subject to six quadratic constraints (the rows must form an orthonormal system). The common approach to this problem is to parameterize the set of rotation matrices by unit quaternions, as follows:$$R=\left(\begin{array}{ccc} q_{1}^{2}+q_{2}^{2}-q_{3}^{2}-q_{4}^{2} & 2q_{2}q_{3}+2q_{1}q_{4} & 2q_{2}q_{4}-2q_{1}q_{3}\\ 2q_{2}q_{3}-2q_{1}q_{4} & q_{1}^{2}+q_{3}^{2}-q_{2}^{2}-q_{4}^{2} & 2q_{3}q_{4}+2q_{1}q_{2}\\ 2q_{1}q_{3}+2q_{2}q_{4} & 2q_{3}q_{4}-2q_{1}q_{2} & q_{1}^{2}+q_{4}^{2}-q_{2}^{2}-q_{3}^{2} \end{array}\right)$$
which transforms the problem into one of minimizing a quartic over the three-dimensional unit sphere $S^{3}=\left\{\left(q_{1},q_{2},q_{3},q_{4}\right)|q_{1}^{2}+q_{2}^{2}+q_{3}^{2}+q_{4}^{2}=1\right\}$. This solution is global and does not rely on iterative schemes. In an extensive set of experiments [5], it was found to be more accurate than other methods. The accuracy of this approach is also noted in many other publications, including the recent [9,10]. Since the running time of the algorithm in [5] is faster than the ones provided in other papers, which approach PnP as the polynomial minimization problem described herewith (more on this in Section 5.1), we directly compared our running times with those given in [5], but the improvement was generic, as will be elaborated in Section 3.

We shall refer to the polynomial that should be minimized as $p_{4}$ and, to reduce equation clutter, will denote its variables as x,y,z,w, and not $q_{1},q_{2},q_{3},q_{4}$.

Due to the great importance of PnP in real-life applications, the minimization problem has been addressed in numerous papers, starting with [4]; for a recent survey, see [5]. There does not exist, however, a fast solution that is guaranteed to work in all problem instances; solving with Lagrange multipliers leads to a set of equations for which no closed-form solution exists. Therefore, it is customary to apply a convex relaxation approach, which we describe in Section 2. In Section 3, we describe a faster solution to the relaxation approach, which also requires less memory. In Section 4, we present a different type of solution, which achieves a guaranteed approximation to the non-relaxed problem; experimental results are presented in Section 5; conclusions are offered in Section 6; some technical details are provided in Appendix A.

## 2. The Lasserre Hierarchy

Our point of departure from previous work is the approach for minimizing $p_{4}\left(x,y,z,w\right)$ on $S^{3}$. We begin by describing the existing approach; see, for example, [5].

The Lasserre hierarchy [12,13,14] is a powerful tool for solving polynomial optimization problems and has found many applications in a variety of fields, including control theory, robotics, and machine learning. One advantage of the Lasserre hierarchy is that it can be implemented using off-the-shelf semidefinite programming solvers, which makes it accessible to a wide range of users. The main drawback of the Lasserre hierarchy is that the computational complexity of the hierarchy grows rapidly with the degree of the polynomials involved, which limits its applicability to problems with a low to moderate degree. Here, we only describe its application to the problem at hand, minimizing $p_{4}\left(x,y,z,w\right)$ over the three-dimensional unit sphere $S^{3}=\left\{\left(x,y,z,w\right)|x^{2}+y^{2}+z^{2}+w^{2}=1\right\}$.

**Definition** **1**([15])**.**
*(2.4): Given a fourth-degree (quartic) polynomial q(x,y,z,w) and an even integer k≥2, define the corresponding k-th degree problem by*
(3)$$\max_{\gamma }\left[q\left(x,y,z,w\right)-\gamma \right]=\phi \left(x,y,z,w\right)\left(1-x^{2}-y^{2}-z^{2}-w^{2}\right)+s\left(x,y,z,w\right)$$
*where ϕ(x,y,z,w) is an arbitrary polynomial of degree k−2 and s(x,y,z,w) a polynomial of degree *k*, which can be expressed as the Sum Of Squares of polynomials (SOS) of degree k/2.*

The advantage of this so-called “SOS relaxation” is that solving for the maximal γ can be reduced to a Semidefinite Programming (SDP) problem, for which there exist efficient numerical algorithms.

Note that, if the polynomial equality:$$q\left(x,y,z,w\right)-\gamma =\phi \left(x,y,z,w\right)\left(1-x^{2}-y^{2}-z^{2}-w^{2}\right)+s\left(x,y,z,w\right)$$
holds, then, since $1-x^{2}-y^{2}-z^{2}-w^{2}=0$ on $S^{3}$ and obviously s(x,y,z,w)≥0 everywhere, clearly, q(x,y,z,w)≥γ on $S^{3}$; hence, γ is a lower bound on the sought minimum. Alas, there is no useful upper bound on the value of *k* for which the resulting γ is equal to the minimum. This problem is exacerbated by the fact that the complexity of the SDP problem rapidly increases with *k*. To see why, observe the following result (for the proof, see [16]).

A polynomial s(x,y,z,w) of even degree *k* in (x,y,z,w) can be expressed as a sum of squares iff it can be written as $vBv^{t}$, where *v* is the vector of monomials in (x,y,z,w) of a total degree not exceeding k/2 and *B* a semidefinite positive matrix. For example, if k=4, we must have that:$$s\left(x,y,z,w\right)=vBv^{t}$$
where *B* is a 15×15 semidefinite positive matrix, and
$$v=(x^{2},xy,xz,xw,y^{2},yz,yw,z^{2},zw,w^{2},x,y,z,w,1)$$

However, while there are 15 monomials of a total degree no more than 2, for the next case (k=6), we must consider all monomials of a degree no more than 3, of which there are 35, which means that the corresponding *B* will be of size 35×35. Generally, there are 4+k/24 such monomials.

In order to obtain reasonable running times, a solution that has been widely adopted by the computer vision community [4,5] is to solve the SOS relaxation for the case of k=4. This immediately raises the following question: Are there PnP instances in which the solution of the fourth-degree problem is different from the correct one (i.e., the global minimum of $p_{4}$)? To the best of our knowledge, this question was first answered (in the affirmative) in two recent papers [11,17]. These cases appear to be very rare, as described in [17]. An exact answer to the question of just *how* rare they are is unknown, as it touches on the very difficult and yet-unsolved problem of the relative measure of SOS polynomials among all positive polynomials, for which only asymptotic results are known [18].

In this paper, we offer two solutions to optimizing $p_{4}\left(x,y,z,w\right)$:**Algorithm 1:** This solves the four-degree problem, but in a much faster way than [2,3,4,5,9,10], by relying on the fact that $p_{4}\left(x,y,z,w\right)$ is homogeneous (i.e., contains only terms of total degree four). While it is not guaranteed to recover the global minimum, neither is the commonly accepted solution described above. In numerous tests run on real PnP data, both solutions gave identical results, with our proposed method being faster by an order of magnitude.**Algorithm 2:** In order to obtain a *guaranteed* approximation to the global minimum, we optimized $p_{4}$ on “slices” of $S^{3}$, each of which is a linear image of the two-dimensional unit sphere $S^{2}$. Then, we applied a famous theorem of Hilbert, which states that every positive homogeneous fourth-degree polynomial in three variables can be represented as a sum of squares. We offer an analysis, backed by experiments, to determine how many slices are required to obtain a good approximation of the global minimum over $S^{3}$.

We now proceed to a detailed study of the relaxed problem and offer our improvement.

## 3. Algorithm 1: Approach and Reduction of Complexity

We began by studying the complexity of the widely used fourth-degree relaxation, sketched in Section 2. Recall the problem:$$\max_{\gamma }\left[p_{4}\left(x,y,z,w\right)-\gamma \right]=\phi \left(x,y,z,w\right)\left(1-x^{2}-y^{2}-z^{2}-w^{2}\right)+s\left(x,y,z,w\right)$$
where
$$s\left(x,y,z,w\right)=vBv^{t},$$
*B* is a 15×15 semidefinite positive matrix:$$v=(x^{2},xy,xz,xw,y^{2},yz,yw,z^{2},zw,w^{2},x,y,z,w,1),$$
and ϕ(x,y,z,w) is an arbitrary second-degree polynomial in x,y,z,w.

This leads to a *Semidefinite Programming* (SDP) problem, in which γ must be maximized, under the following constraints:*B* is a semidefinite positive matrix.There are 70 equalities that must be satisfied, which correspond to the 70 coefficients of $p_{4}\left(x,y,z,w\right)$.These 70 equalities also depend on the 15 coefficients of the quadratic polynomial ϕ(x,y,z,w).

In the SDP formulation, this problem is posed as follows:$$\mathrm{Minimize}\;\;C\cdot X\;\;\mathrm{subject}\;\mathrm{to}\;\;X⪰0\;\;\mathrm{and}\;\;X\cdot A_{i}=b_{i}$$
where, for two matrices of the same dimensions A,B, A·B is defined as $\sum_{i,j}A_{i,j}B_{i,j}$ (also equal to the trace (*AB*)) and for a square and symmetric matrix *P*, P⪰0 stands for *P* being semidefinite positive.

This problem is typically solved by formulating the following dual-problem, which also allows recovering the point at which the minimum is obtained:$$\mathrm{Maximize}\;\;\sum_{i}b_{i}y_{i}\;\;\mathrm{subject}\;\mathrm{to}\;\;C-\sum_{i}y_{i}A_{i}⪰0$$

The SDP problem can be solved using available packages such as SDPA [19] or by direct optimization [5].

### 3.1. Improving Algorithm 1

The size of the SDP problem is, naturally, a crucial factor in the complexity of its solution. In the above problem, the matrix *X* is of size 15×15, and there are 70 equality constraints, corresponding to the 70 coefficients of $p_{4}\left(x,y,z,w\right)$. In addition, there are 15 “free” (i.e., without constraints) coefficients of ϕ(x,y,z,w), which must be recovered.

Proceeding as in [19], this entails an optimization procedure based on the famous *iterative barrier method*, with a Hessian of size 70×70 and, with each iteration step, solving linear systems of size 85×85.

We propose a substantially simpler solution, which is an order of magnitude faster than previous work, while producing identical results in numerous tests on real PnP data. Specifically, in our solution, the matrix *X* is smaller (10×10 vs. 15×15), and there are fewer coefficients to recover (34 vs. 85). Since a major part of the solution is computing the inverse of the Hessian matrix (which is $O(n^{3})$ for a matrix of size n×n) and our Hessian is quite smaller, the proposed solution is about 10-times faster than in previous work (it takes about 1 ms, on a relatively weak i7-6500u processor to find the minimum).

#### Details of Our Proposed Improvement

Our proposed solution uses the fact that the polynomial $p_{4}\left(x,y,z,w\right)$ is *homogeneous*, meaning it contains only monomials with total degree four, and all the other coefficients are equal to 0.

Suppose that the minimum of $p_{4}\left(x,y,z,w\right)$ on $S^{3}$ is γ. Since on $S^{3}$, it holds that $x^{2}+y^{2}+z^{2}+w^{2}=1$, then on $S^{3}$, the following trivially holds:$$q_{4}\left(x,y,z,w\right)≜p_{4}\left(x,y,z,w\right)-\gamma {\left(x^{2}+y^{2}+z^{2}+w^{2}\right)}^{2}\geq 0$$

However, $q_{4}\left(x,y,z,w\right)$ is also homogeneous, as it clearly contains only monomials of total degree four; hence we have, for every real number α:$$q_{4}\left(\alpha x,\alpha y,\alpha z,\alpha w\right)=\alpha^{4}q_{4}\left(x,y,z,w\right)$$
but for every $\left(x,y,z,w\right)\in R^{4}$, $\frac{\left(x,y,z,w\right)}{∥(x,y,z,w)∥}\in S^{3}$, and
$$q_{4}\left(x,y,z,w\right)={∥\left(x,y,z,w\right)∥}^{4}\;q_{4}\left(\frac{\left(x,y,z,w\right)}{∥(x,y,z,w)∥}\right);$$
therefore, $q_{4}\left(x,y,z,w\right)\geq 0$ on $S^{3}$ iff $q_{4}\left(x,y,z,w\right)\geq 0$ on $R^{4}$.

Next, we followed the standard SOS relaxation [12]: assume that if a fourth-degree polynomial $q_{x}\left(x,y,z,w\right)$ is non-negative on $R^{4}$, it can be written as
(4)$$q_{4}\left(x,y,z,w\right)=\left(x^{2},xy,\ldots ,w^{2}\right)B{\left(x^{2},xy,\ldots ,w^{2}\right)}^{t}$$
for some 10×10 matrix *B* such that B⪰0. Note that we used the fact that the polynomial is homogeneous; hence, the vector of monomials contains only 10 entries, and not 15.

Therefore, we can replace the previous optimization problem by
(5)$$\;\;\;\;\max\gamma \;\;\mathrm{such}\;\mathrm{that}\;\;p_{4}\left(x,y,z,w\right)-\gamma {\left(x^{2}+y^{2}+z^{2}+w^{2}\right)}^{2}=\left(x^{2},xy,\ldots ,w^{2}\right)B{\left(x^{2},xy,\ldots ,w^{2}\right)}^{t}$$

The size of the problem can be further reduced as follows. By equating the coefficient of $x^{4}$ on both sides of Equation (Equation 5) and denoting the coefficient of $p_{4}\left(x,y,z,w\right)$ by $a_{4000}$, we can see that $a_{4000}-\gamma =B_{1,1}$. Since we wish to maximize γ, we can minimize $B_{1,1}$ (because $a_{4000}$ is fixed). Therefore, now, the problem becomes
$$\begin{array}{ccc} & & \min B_{1,1}\;\;\mathrm{such}\;\mathrm{that}\;\;\;p_{4}\left(x,y,z,w\right)-\left(a_{4000}-B_{1,1}\right){\left(x^{2}+y^{2}+z^{2}+w^{2}\right)}^{2}=\\ & & \left(x^{2},xy,\ldots ,w^{2}\right)B{\left(x^{2},xy,\ldots ,w^{2}\right)}^{t} \end{array}$$

This problem has no inequality constraints and can be described as the problem of minimizing trace(CB), where *C* is the 10×10 matrix with $C_{1,1}=1$ and all other elements equal to 0, under 34 equality constraints on *B*, corresponding to the 34 coefficients (we do not need one for $B_{1,1}$). In addition to working with smaller matrices, we do not require any auxiliary variables as the 15 coefficients of ϕ(x,y,z,w) in previous work, e.g., [5], and the constraint matrices corresponding to the 34 coefficients are very sparse (see Appendix A), which further reduces the running time [20].

Next, we considered the dual-problem, with the well-known method of replacing
$$\min\;\mathrm{trace}\left(CB\right)\;\;\mathrm{such}\;\mathrm{that}\;\;\mathrm{trace}\left(A_{i}B\right)=b_{i}$$
with
$$\max\;\;b_{1}y_{1}+\ldots +b_{k}y_{k}\;\;\mathrm{such}\;\mathrm{that}\;\;C-b_{1}A_{1}-\ldots -b_{k}A_{k}\;\mathrm{is}\;a\;\mathrm{semidefinite}\;\mathrm{positive}\;\mathrm{matrix},$$
yielding the following problem:$$\begin{array}{ccc} & & \mathrm{Maximize}\;\;y_{34}a_{3100}+y_{33}a_{3010}+y_{11}a_{0211}+y_{12}\left(a_{0220}-2a_{4000}\right)+y_{13}a_{0301}+y_{14}a_{0310}\\ & & +y_{15}\left(a_{0400}-a_{4000}\right)+y_{16}a_{1003}+y_{17}a_{1012}+y_{18}a_{1021}+y_{19}a_{1030}+y_{32}a_{3001}+\\ & & y_{31}\left(a_{2200}-2a_{4000}\right)+y_{30}a_{2110}+y_{29}a_{2101}+y_{20}a_{1102}+y_{28}\left(a_{2020}-2a_{4000}\right)+y_{21}a_{1111}+\\ & & y_{22}a_{1120}+y_{23}a_{1201}+y_{24}a_{1210}+y_{25}a_{1300}+y_{26}\left(a_{2002}-2a_{4000}\right)+y_{27}a_{2011}+\\ & & y_{10}\left(a_{0202}-2a_{4000}\right)+y_{9}a_{0130}+y_{8}a_{0121}+y_{7}a_{0112}+y_{6}a_{0103}+y_{5}\left(a_{0040}-a_{4000}\right)+\\ & & y_{4}a_{0031}+y_{1}\left(a_{0004}-a_{4000}\right)+y_{2}a_{0013}+y_{3}\left(a_{0022}-2a_{4000}\right) \end{array}$$
such that:$$Y=\left(\begin{array}{cccccccccc} Y_{11} & -y_{34} & -y_{33} & -y_{32} & -y_{31} & -y_{30} & -y_{29} & -y_{28} & -y_{27} & -y_{26}\\ -y_{34} & -y_{31} & -y_{30} & -y_{29} & -y_{25} & -y_{24} & -y_{23} & -y_{22} & -y_{21} & -y_{20}\\ -y_{33} & -y_{30} & -y_{28} & -y_{27} & -y_{24} & -y_{22} & -y_{21} & -y_{19} & -y_{18} & -y_{17}\\ -y_{32} & -y_{29} & -y_{27} & -y_{26} & -y_{23} & -y_{21} & -y_{20} & -y_{18} & -y_{17} & -y_{16}\\ -y_{31} & -y_{25} & -y_{24} & -y_{23} & -y_{15} & -y_{14} & -y_{13} & -y_{12} & -y_{11} & -y_{10}\\ -y_{30} & -y_{24} & -y_{22} & -y_{21} & -y_{14} & -y_{12} & -y_{11} & -y_{9} & -y_{8} & -y_{7}\\ -y_{29} & -y_{23} & -y_{21} & -y_{20} & -y_{13} & -y_{11} & -y_{10} & -y_{8} & -y_{7} & -y_{6}\\ -y_{28} & -y_{22} & -y_{19} & -y_{18} & -y_{12} & -y_{9} & -y_{8} & -y_{5} & -y_{4} & -y_{3}\\ -y_{27} & -y_{21} & -y_{18} & -y_{17} & -y_{11} & -y_{8} & -y_{7} & -y_{4} & -y_{3} & -y_{2}\\ -y_{26} & -y_{20} & -y_{17} & -y_{16} & -y_{10} & -y_{7} & -y_{6} & -y_{3} & -y_{2} & -y_{1} \end{array}\right)$$
where $Y_{11}=1+2y_{26}+y_{28}+y_{1}+2y_{31}+2y_{3}+y_{5}+2y_{10}+2y_{12}+y_{15}$ is a semidefinite positive matrix.

### 3.2. Finding a Good Starting Point

It is well known that finding a good initial point greatly affects the running time of SDP algorithms. These algorithms seek a solution in which a matrix *B* must be semidefinite positive. Typically, they follow the “barrier method”, which imposes a penalty on matrices whose determinant is very small, thus keeping the candidate matrix from crossing the “barrier” defined by the set of singular matrices [14,19,21]. Ideally, an initial point is at the center of the region that contains the viable solutions for *B*. We next describe how this can be achieved for our problem.

Recall that, in the proposed solution, the term containing the matrix *B* equals
$$\left(x^{2},xy,\ldots ,w^{2}\right)B{\left(x^{2},xy,\ldots ,w^{2}\right)}^{t}.$$

It is known from the theory of SDP [12,14] that the optimal *B* consists of the corresponding *moments* of the solution $\left(x_{0},y_{0},z_{0},w_{0}\right)$ (and this is what allows extracting the solution from *B*). Since we know that all points $\left(x_{0},y_{0},z_{0},w_{0}\right)\in S^{3}$ are equally viable as solutions, we can compute the center of mass of the viable *B*-region by
$$\begin{array}{ccc} & & B_{1,1}=\int_{S^{3}}x^{4}dxdydzdw\\ & & B_{1,2}=\int_{S^{3}}x^{3}ydxdydzdw\\ & & \vdots \\ & & B_{10,10}=\int_{S^{3}}w^{4}dxdydzdw \end{array}$$

These integrals can be computed as follows. Firstly, note that, from the symmetry and parity considerations, all of them are equal to 0, except
$$\int_{S^{3}}x^{4}dxdydzdw,\;\;\;\int_{S^{3}}x^{2}y^{2}dxdydzdw\ldots ,$$
i.e., all those containing a fourth power of a variable or the product of squares of two variables. Next, we computed $\int_{S^{3}}x^{4}dxdydzdw$ using the well-known parameterization of $S^{3}$:$$\begin{array}{cc} x & =\cos(\phi_{1})\\ y & =\sin\left(\phi_{1}\right)\cos\left(\phi_{2}\right)\\ z & =\sin\left(\phi_{1}\right)\sin\left(\phi_{2}\right)\cos\left(\phi_{3}\right)\\ w & =\sin\left(\phi_{1}\right)\sin\left(\phi_{2}\right)\sin\left(\phi_{3}\right) \end{array}$$
and the Jacobian, corresponding to the three-dimensional area element, equals $\sin^{2}\left(\phi_{1}\right)\sin\left(\phi_{2}\right)$. Therefore, after normalizing by the surface area of $S^{3}$, which equals $2\pi^{2}$, we obtain
$$\frac{1}{2\pi^{2}}\int_{0}^{\pi }\int_{0}^{\pi }\int_{0}^{2\pi }\cos^{4}\left(\phi_{1}\right)\sin^{2}\left(\phi_{1}\right)\sin\left(\phi_{2}\right)d\phi_{1}d\phi_{2}d\phi_{3}=\frac{1}{8}$$
and similarly, the matrix elements corresponding to $x^{2}y^{2}$ equal $\frac{1}{24}$. It follows that the center of mass of the viable region, henceforth denoted by $C_{m}$, is equal to:$$\frac{1}{24}\left(\begin{array}{cccccccccc} 3 & 0 & 0 & 0 & 1 & 0 & 0 & 1 & 0 & 1\\ 0 & 1 & 0 & 0 & 0 & 0 & 0 & 0 & 0 & 0\\ 0 & 0 & 1 & 0 & 0 & 0 & 0 & 0 & 0 & 0\\ 0 & 0 & 0 & 1 & 0 & 0 & 0 & 0 & 0 & 0\\ 1 & 0 & 0 & 0 & 3 & 0 & 0 & 1 & 0 & 1\\ 0 & 0 & 0 & 0 & 0 & 1 & 0 & 0 & 0 & 0\\ 0 & 0 & 0 & 0 & 0 & 0 & 1 & 0 & 0 & 0\\ 1 & 0 & 0 & 0 & 1 & 0 & 0 & 3 & 0 & 1\\ 0 & 0 & 0 & 0 & 0 & 0 & 0 & 0 & 1 & 0\\ 1 & 0 & 0 & 0 & 1 & 0 & 0 & 1 & 0 & 3 \end{array}\right)$$

After offering our improvement for the solution of the relaxed problem, we next handled the non-relaxed version and offer a fast, easily parallelizable solution with guaranteed accuracy.

## 4. Algorithm 2: Solution with “Slices”

In this section, we offer our additional contribution to solving the algebraic optimization problem. While efficient and widely used, the method we discussed in Section 2 and Section 3 does not solve the original problem of minimizing $p_{4}\left(x,y,z,w\right)$ on the unit sphere $S^{3}$, but a relaxed version of it; specifically, the relaxation consists of replacing the condition that a polynomial is everywhere positive (which is notoriously difficult to check; actually, it is NP-complete [12,14]), by an easy-to-verify (and impose) condition: that the polynomial is the sum of squares of polynomials—or, equivalently, that it can be written as $vBv^{t}$, where *B* is a semidefinite positive matrix and *v* the vector of monomials with half the degree of the given polynomial (clearly an odd degree polynomial cannot be everywhere positive).

Since the entire well-developed and very successful theory of Sum Of Squares (SOS) optimization relies on this relaxation, a great deal of effort has been put into studying the following question: How “tight” is the relaxation, i.e., can the difference between the optimal solution and the relaxed one be bounded? There are still no general answers to this question. In the context of the PnP problem, recent work showed that, in principle, the relaxation may fail [17], and in [11], a concrete example was provided for such a failure, even in the six-degree stage of the Lasserre hierarchy (Definition 1).

We propose a very simple approximation algorithm to optimizing $p_{4}\left(x,y,z,w\right)$, with the following properties:$p_{4}\left(x,y,z,w\right)$ is optimized on a pre-set number of “slices” of $S^{3}$, defined by the intersection of $S^{3}$ with sub-spaces defined by $w=\beta_{i}z,i=-n,\ldots ,n$.Applying a famous theorem of Hilbert [22] (see the ensuing discussion) to our approach described in Section 3, it turns out that the absolute minimum on every slice can be *exactly* computed using the relaxation we presented in Section 3.The optimization for each slice is extremely fast, as the sought positive semidefinite matrices are of size 6×6.Since the difference between the coefficients of the polynomials of two nearby slices is very small, the optimization solution for the *i*th slice can be used to find a good starting point for optimizing over the (i + 1)th slice.Further speedup of the optimization can be achieved by parallelizing the optimization, by diving it into separate optimizations run in parallel over sets of adjacent slices.

We next describe the method. First, for a given scalar β, denote
$$S_{\beta }=S^{3}⋂\left\{\left(x,y,z,w\right)|w=\beta z\right\}$$
which we refer to as the β-*slice*. In Figure 2, the equivalent notion of slices of $S^{2}$ is depicted.

Next, we address the problem of optimizing $p_{4}\left(x,y,z,w\right)$ on $S_{\beta }$. Clearly,
$$p_{4}\left(x,y,z,\beta z\right)≜p_{\beta }\left(x,y,z\right)$$
is a fourth-degree homogeneous polynomial in three variables, and by re-scaling *z*, it can be considered as defined on $S^{2}=\left\{\left(x,y,z\right)|x^{2}+y^{2}+z^{2}=1\right\}$. As in Algorithm 1, we can seek its minimum by solving
$$\begin{array}{c} \;\;\;\;\max\gamma \;\;\mathrm{such}\;\mathrm{that}\;\;\;p_{4}\left(x,y,z\right)-\gamma {\left(x^{2}+y^{2}+z^{2}\right)}^{2}=\left(x^{2},xy\ldots z^{2}\right)B{\left(x^{2},xy\ldots z^{2}\right)}^{t} \end{array}$$

However, due to the following famous theorem of Hilbert, there is a crucial difference between this problem and that of Algorithm 1.

**Theorem** **1.**
*Every non-negative homogeneous polynomial of degree four in three variables can be expressed as a sum of squares or, equivalently, as*

$$\left(x^{2},xy,xz,y^{2},yz,z^{2}\right)B{\left(x^{2},xy,xz,y^{2},yz,z^{2}\right)}^{t}$$

*for some 6×6 semidefinite positive matrix B.*


The theorem was first proven by Hilbert [22]. Easier-to-follow proofs were later discovered, for example Chapter 4 in [16], but they are still quite elaborate. It follows immediately that the corresponding SDP problem will always return the global minimum over $S_{\beta }$.

### Approximation Quality of the Slices Method

In order to estimate how well the minimum over the slices approximates the minimum over $S^{3}$, we first provide a result that quantifies how well the slices approximate every point on $S^{3}$.

**Lemma** **1.**
*It is possible, with n slices, to approximate every point on $S^{3}$ to within a distance of $O\left(\frac{1}{n}\right)$.*


**Proof.** We shall use two types of slices: w=βz and z=βw. Hence, we can assume without loss of generality that |w|≤|z|. We used slices with $\beta_{i}=-1,-\frac{n-1}{n},\ldots ,0,\frac{1}{n},\ldots ,1$. Using rotational symmetry, we can assume that w=βz, for $0\leq \beta \leq \frac{1}{n}$, and sought the closest point to (x,y,z,w) on the hyperplane defined by w=0. This point is readily seen to equal
$$\frac{\left(x,y,z\right)}{\sqrt{x^{2}+y^{2}+z^{2}}}$$
and its squared distance from (x,y,z,w) equals
$${\left(x-\frac{x}{\sqrt{x^{2}+y^{2}+z^{2}}}\right)}^{2}+{\left(y-\frac{y}{\sqrt{x^{2}+y^{2}+z^{2}}}\right)}^{2}+{\left(z-\frac{z}{\sqrt{x^{2}+y^{2}+z^{2}}}\right)}^{2}+\beta^{2}z^{2}$$Since we have, for the β-slice, $x^{2}+y^{2}+z^{2}+\beta^{2}z^{2}=1$, the previous expression equals
$${\left(x-\frac{x}{\sqrt{-\beta^{2}z^{2}+1}}\right)}^{2}+{\left(y-\frac{y}{\sqrt{-\beta^{2}z^{2}+1}}\right)}^{2}+{\left(z-\frac{z}{\sqrt{-\beta^{2}z^{2}+1}}\right)}^{2}+\beta^{2}z^{2};$$
its Taylor expansion around β=0 equals
$$\beta^{2}z^{2}+\left(\frac{1}{4}x^{2}z^{4}+\frac{1}{4}y^{2}z^{4}+\frac{1}{4}z^{6}\right)\beta^{4},$$
and since $0\leq \beta \leq \frac{1}{n}$ and |z|≤1, the result immediately follows. □

Next, we analyzed the distance between the minimum over the slices and the minimum over $S^{3}$. Recall that the function to be minimized is:(6)$$\left(x^{2},xy,xz,xw,y^{2},yz,yw,z^{2},zw,w^{2}\right)B\left(\begin{array}{c} x^{2}\\ xy\\ xz\\ xw\\ y^{2}\\ yz\\ yw\\ z^{2}\\ zw\\ w^{2} \end{array}\right),$$
for some semidefinite positive matrix *B*, which depends on the input to the PnP problem (i.e., points $p_{i}$ and directions $u_{i}$; see Section 1). Therefore, we need to analyze the difference between Equation (Equation 6) for a point $p≜\left(x,y,z,w\right)\in S^{3}$ and a point *q* whose distance from *p* is at most $\frac{1}{n}$. Proceeding as in matrix perturbation theory [23], we denote q=p+ε, where $\varepsilon =(\varepsilon_{x},\varepsilon_{y},\varepsilon_{z},\varepsilon_{w})$ denotes a “small” vector, in the sense that all squares of its components are very small and can be ignored in the forthcoming analysis.

We start by estimating the difference between *P* and *Q*, which denotes the length-10 vectors constructed from p,q, i.e., $P=(x^{2},xy,xz,xw,y^{2},yz,yw,z^{2},zw,w^{2})$ and similarly for *Q*. Since a direct computation shows that:$${\left(x^{2}+y^{2}+z^{2}+w^{2}\right)}^{2}-{∥P∥}^{2}=x^{2}w^{2}+y^{2}w^{2}+z^{2}w^{2}+x^{2}y^{2}+x^{2}z^{2}+y^{2}z^{2},$$
it is clear that ∥P∥≤1. Furthermore, we have:$${∥P-Q∥}^{2}={\left({\left(x+\varepsilon_{x}\right)}^{2}-x^{2}\right)}^{2}+{\left(\left(x+\varepsilon_{x}\right)\left(y+\varepsilon_{y}\right)-xy\right)}^{2}+\cdots +{\left({\left(w+\varepsilon_{w}\right)}^{2}-w^{2}\right)}^{2}.$$

Ignoring terms of order at least two in ε, we obtain:$$4\left(x^{2}\varepsilon_{x}^{2}+\ldots +w^{2}\varepsilon_{w}^{2}\right)+{\left(x\varepsilon_{y}+y\varepsilon_{x}\right)}^{2}+\ldots +{\left(z\varepsilon_{w}+w\varepsilon_{z}\right)}^{2}.$$

Expanding and using $x^{2}+y^{2}+z^{2}+w^{2}=1$, as well as the Cauchy–Schwarz inequality allow bounding the last expression by $16\varepsilon^{2}$; hence, we have $∥P-Q∥\leq 4∥\varepsilon ∥\leq \frac{4}{n}$. Lastly, denoting Q=P+E and again ignoring the higher-order terms, we obtain (using the inequality ∥P∥≤1):$$PBP^{t}-QBQ^{t}\approx 2PBE^{t}\leq 2∥B∥∥E∥\leq \frac{8∥B∥}{n}\;\;\left(∥B∥=\mathrm{operator}\;\mathrm{norm}\;\mathrm{of}\;\;B\right),$$
which provides a bound on the error of the approximate minimum.

Next, the results of both our algorithms are presented.

## 5. Results


Experimental Data


The data we used for comparison with [5], as well as the results for the various methods tested in [5] can be found at https://tinyurl.com/bdudy3z3, accessed on 11 April 2023. These data are described in detail in [5] and consist of webcam and drone data, as well as noisy and synthetic data. Altogether, there are 230MB of data, consisting of image coordinates $p_{i}$ and line of sight direction $u_{i}$ (Equation (Equation 1)). The results are for all methods and consist of the rotation matrix *R* and translation *T* between frames (Equation (Equation 1)). Our results were identical to those of the NPnP method introduced in [5], but about ten-times faster on average (Figure 3).

### 5.1. Experimental Results for Algorithm 1

We chose the experiments in the recent paper [5] as a baseline for comparison, for the following reasons:It contains extensive experiments, comparing many different algorithms, on different types of data, also with varying levels of noise.To the best of our knowledge, it is the only paper that addresses the solution of PnP as an SDP problem (first suggested in [4]) with dedicated software, as opposed to an off-the-shelf SDP package. The considerable effort to tailor the solution to PnP reduced the running time by a factor of about three relative to standard packages.The comparison of the global optimization approach to PnP (which we also followed here) proved it to be more accurate than other popular methods, including EPnP [2], DLS [24], RPnP [25], and the iterative method presented in [26].Additional evidence for the superior performance (in terms of accuracy) of SDP-based methods is provided in other papers, including the recent [9]; this further motivated our effort to reduce the running time of SDP.

Using the approach described in Section 3.1 with $C_{m}$ (Section 3.2) as an initial point for the SDP iterations, with the SDPA package [19], yielded a reduction in the running time of about an order of magnitude relative to the state-of-the-art results in [5]. This improvement can be expected, given the reduction in the number of variables relative to previous work, which entails a reduction in the size of matrices that should be inverted (from 85×85 to 34×34). Thus, its advantage does not depend on the specific SDP tools applied, nor the computational platform.

The proposed algorithm was compared to [5] on 10,000 PnP inputs. In all cases, the results were identical up to four decimal points. The distributions of the running times are presented in Figure 3.

### 5.2. Experimental Results for Algorithm 2

In experiments on 10,000 instances of real PnP data, the error was quite small and decreased rapidly when the number of slices was increased. Table 1 provides the average and maximal relative error of Algorithm 2 vis-à-vis the global minimum (note that the error was *one-sided*, that is the global minimum was always smaller than the one provided by Algorithm 2).


Running Time of Algorithm 2


For an individual slice, the solution consisted of optimizing a quartic ternary form, i.e., a homogeneous polynomial of degree four with three variables. Using the approach described in Section 3 for the similar problem with four variables led to an SDP problem with a 6×6 matrix, which can be solved very quickly. Further speedup was obtained by partitioning the slices into disjoint sets, each consisting of consecutive β values, and solving for each set separately, in parallel, on a standard multi-core processor. Since the coefficients for polynomials of consecutive slices are nearly identical, further speedup can be obtained by using the solution for the previous slice as a starting point for the current one; this is, however, not trivial as for other optimization problems, as we now elaborate.


Using an Initial Guess for Consecutive Slices


Suppose we have optimized the polynomial $p_{\beta_{1}}\left(x,y,z\right)$, and wish to continue with the consecutive slice, that is optimize $p_{\beta_{2}}\left(x,y,z\right)$, where $\beta_{2}$ is very close to $\beta_{1}$. Since the coefficients of both polynomials are very close, ostensibly, we could use the solution for $p_{\beta_{1}}\left(x,y,z\right)$ as an initial point for optimizing $p_{\beta_{2}}\left(x,y,z\right)$; alas, there is a subtle issue at play. Recall that the sought semidefinite positive matrix *B* appears in the expression, which we optimized as
(7)$$\left(x^{2},xy,xz,y^{2},yz,z^{2}\right)B\left(\begin{array}{c} x^{2}\\ xy\\ xz\\ y^{2}\\ yz\\ z^{2} \end{array}\right);$$
therefore, it is well known from moment theory [14] that, if $\left(x_{0},y_{0},z_{0}\right)$ is the point at which the minimum is attained, the optimal *B* is of the form:$$\left(\begin{array}{cccccc} x_{0}^{4} & x_{0}^{3}y_{0} & x_{0}^{3}z_{0} & x_{0}^{2}y_{0}^{2} & x_{0}^{2}y_{0}z_{0} & x_{0}^{2}z_{0}^{2}\\ x_{0}^{3}y_{0} & x_{0}^{2}y_{0}^{2} & x_{0}^{2}y_{0}z_{0} & x_{0}\;y_{0}^{3} & x_{0}\;y_{0}^{2}z_{0} & x_{0}y_{0}\;z_{0}^{2}\\ x_{0}^{3}z_{0} & x_{0}^{2}y_{0}z_{0} & x_{0}^{2}z_{0}^{2} & x_{0}\;y_{0}^{2}z_{0} & x_{0}y_{0}\;z_{0}^{2} & x_{0}\;z_{0}^{3}\\ x_{0}^{2}y_{0}^{2} & x_{0}\;y_{0}^{3} & x_{0}\;y_{0}^{2}z_{0} & y_{0}^{4} & y_{0}^{3}z_{0} & y_{0}^{2}z_{0}^{2}\\ x_{0}^{2}y_{0}z_{0} & x_{0}\;y_{0}^{2}z_{0} & x_{0}y_{0}\;z_{0}^{2} & y_{0}^{3}z_{0} & y_{0}^{2}z_{0}^{2} & y_{0}\;z_{0}^{3}\\ x_{0}^{2}z_{0}^{2} & x_{0}y_{0}\;z_{0}^{2} & x_{0}\;z_{0}^{3} & y_{0}^{2}z_{0}^{2} & y_{0}\;z_{0}^{3} & z_{0}^{4} \end{array}\right).$$

Evidently, this matrix is highly singular (of rank one), while a good starting point should be in the interior of the feasible region (semidefinite positive matrices). We tested two solutions to this problem, which still allow gaining from the proximity of the two slices:While solving for the first slice in a set of nearby slices, if *n* iterations are required (for the SDPA package we used, typically n=14), store the $\frac{n}{2}$ iteration, and use it as a starting point for the other slices. This initial guess, while closer to the solutions than a random starting point, is well within the interior.As in Section 3, compute the center of the feasible region, $C_{m}$, for our problem. Then, if the solution matrix for the $\beta_{1}$ slice is $B_{\beta_{1}}$, choose as an initial point for the $\beta_{2}$ slice ($\beta_{2}\approx \beta_{1}$) a convex combination of $C_{m}$ and $B_{\beta_{1}}$. Intuitively speaking, we take the (singular) solution for $\beta_{1}$ and slightly “push it away” from the boundary of the feasible region, thus creating an initial point that is both close to the solution and in the interior of the feasible region.

Method 2 above produced slightly better results in terms of running times. For 40 slices, the average running time was 1.35 ms, i.e., slightly higher than Algorithm 1.

## 6. Conclusions and Future Work

The PnP problem (recovering camera translation and rotation from point matches) is fundamental in three-dimensional computer vision. A widely used and accurate solution requires minimizing a quartic polynomial over $S^{3}$. Alas, this minimization problem is difficult; therefore, a very common solution is to apply convex relaxation, which reduces the solution to that of Semidefinite Programming (SDP).

While, typically, the solution of the relaxed problem is identical to that of the original problem, PnP configurations for which the relaxation fails were provided recently [11,17].

In this paper, we presented two novel algorithms:Algorithm 1 solves the relaxed problem, but is faster than the state-of-the-art solutions, as it optimizes over smaller matrices (10×10 vs. 15×15) and contains fewer variables (34 vs. 85), thus reducing the running time by roughly an order of magnitude.Algorithm 2 provides a fast, *guaranteed* approximation to the original (non-relaxed) PnP problem, by solving it over a set of “slices” through $S^{3}$. This solution makes use of a famous theorem of Hilbert, to prove that the solution for each slice is optimal. We provided a bound for the difference of the minimum over the slices vis-à-vis the minimum over $S^{3}$ and presented methods to speed up the optimization.

Future work will consider problems with more variables, for example simultaneous recovery of a few rotation matrices [17].

## Figures and Tables

**Figure 1 sensors-23-05585-f001:**
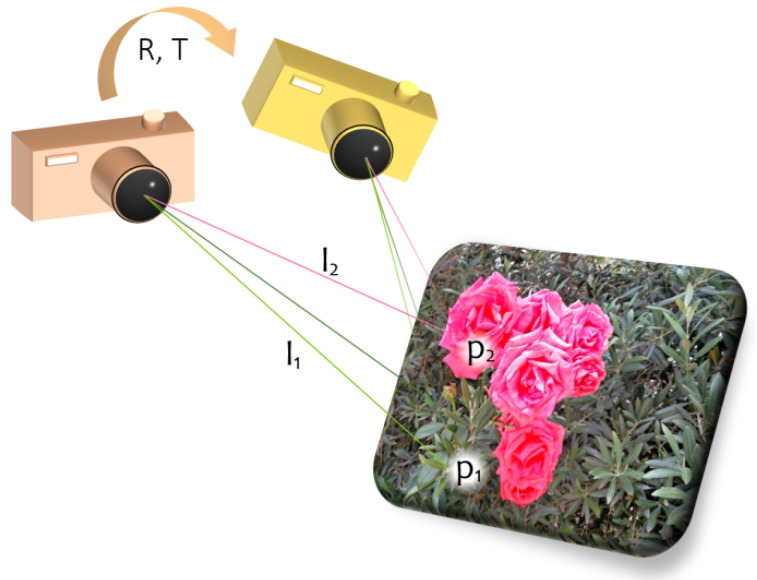
Geometric interpretation of the PnP problem.

**Figure 2 sensors-23-05585-f002:**
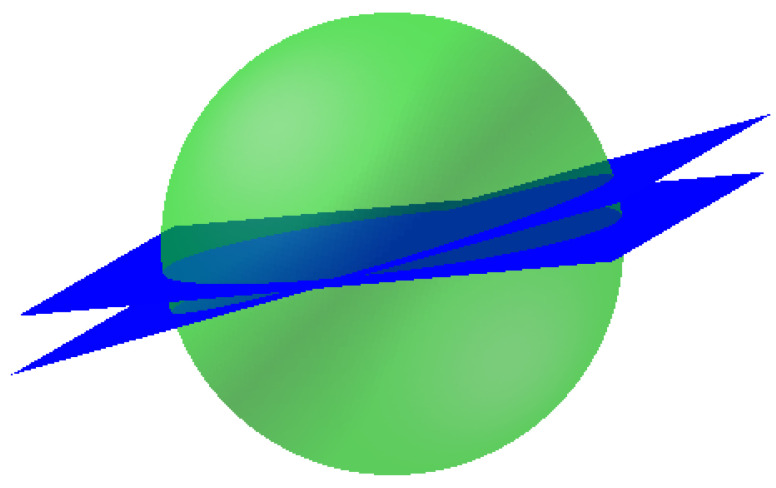
“Slices” through the two-dimensional sphere $S^{2}$.

**Figure 3 sensors-23-05585-f003:**
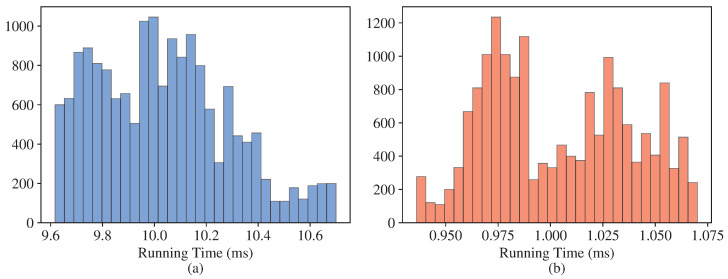
Histograms of running times (milliseconds) for the state-of-the-art algorithm in [5] (**a**) and the proposed algorithm (**b**).

**Table 1 sensors-23-05585-t001:** Relative error, Algorithm 2.

Number of Slices	Average Error	Maximal Error
50	0.0014	0.0022
100	0.00023	0.00033
200	0.000061	0.00012
400	0.000017	0.000028

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
