# Peer review of "A Fast and Reliable Solution to PnP, Using Polynomial Homogeneity and a Theorem of Hilbert"

_sensors, 2023, doi:10.3390/s23125585_

Round 1

Reviewer 1 Report

 The manuscript contains several grammatical errors and typos. I asked the authors to revise the manuscript using any tool.

Author Response

We are very grateful to the reviewers for their detailed and insightful
comments, and have revised the paper accordingly. Here are our replies:

Reviewer 1
----------

1.1: "Slight history of Perspective-n-Point problem which may include..."

Added, please see Introduction. Naturally, it is impossible to cover the
entire rich history of PnP in anything but a long survey paper; so we
have followed the advice of the reviewer, and also, especially concentrated
on the relevant line of research. Also, we compared our approach to recent
work.

1.2: "Also, the introduction must show which gap the paper will fill
in the literature."

Done, we have emphasized the position of our work relative to the research
on PnP, and its contribution.

1.3: "Also, it should be amended to cover the main motivation of the work."

We better emphasized the contribution, focusing on the improvement of the
proposed approach, and better delineated the contributions to the relaxed
and non-relaxed problems.

1.4: "Additionally, at the end of this section, an organization of the paper
should be provided."

Done, this was added to the Introduction.

1.5: "A brief paragraph about the Lasserre hierarchy:"

Was added.

1.6: "Definition 1 needs a citing reference."

Added.

1.7: "Subsection 3.1 includes one subsection 3.1.1. Why?!."

Corrected.

1.8: "I suggest merging two subsections 3.3 and 3.4."

Done; further, both were moved to the new Section 5 (Results) at the
request of Reviewer 2.

1.9: "Subsection 4.3 includes one subsection 4.3.1. Why?!"

Corrected.

1.10 :"I suggest merging two subsections 4.1 and 4.2."

Done; further, both were moved to the new Section 5 (Results) at the
request of Reviewer 2.

1.11: "Appendix I must be without numbers."

Corrected.

1.12: "The authors should organize the flow of the manuscript by adding
brief texts before and after the sections."

Done for the relevant sections.

1.13: "The manuscript contains several grammatical errors and typos. I
asked the authors to revise the manuscript using any tool."

We did our best, with the help of a professional, to correct all typos
and grammatical errors.

Reviewer 2 Report

The authors present a fast and reliable solution to PNP, using polynomial
homogeneity and a theorem of Hilbert. PNP is one of the most extensively studied problems in 3D computer vision and there exist several comprehensive works on it. The authors claim to propose an algorithm which is 10x faster than the state of the art work and a fast approximation using Hilbert.

I have several concerns with the proposed approach which are listed below:

1. Most of the equations used in this work are known. They should be properly cited. Any change in any equation or a newly proposed equation should be clearly mentioned as "our proposed work" so that the readers can differentiate between the contributions of the work and the existing work.

2. Literature review is very weak and does not cover the domain in depth and breadth. PNP is an extensively studied problem and one can find hundreds of related works. However, the authors haven't covered the literature in a meaningful manner.

3. The proposed approach heavily relies on the existing work [2] (section 2). I am not able to find the novelty of the proposed work. What are the main contributions? Section 3.1.1 discusses the proposed improvement, however, the novelty of the work is still questionable.

4. The proposed approach is not tested on publicily available datasets. The proposed approach heavily relies on [2] and the authors also choose the same paper as benchmark. They also use the dataset provided by [2]. The data, which is 230MBs, is not sufficient and the findings on such a small dataset can be used to claim that the proposed work is "10x faster and reliable". The proposed work is 10x faster than [2] and only on the dataset provided by [2].

5. Authors must test their proposed work on larger, extensive and heterogenous publicly available datasets and also compare work with other related works (not with [2] only).

6. One expects other evaluation parameters as well e.g. computation time, time complexity of the algorithm, space complexity etc. Authors should also include specs of the system on which the algorithm was tested.

7. Section 4 should be merged with section 3 and the name should be proposed work. Similarly all of the results should be in the results section. Currently, the paper is in a very odd format.

8. Section 4.2: What dataset was used? Have you compared your findings with any of the existing works? The claim "guaranteed" is not supported anywhere in the manuscript.

Minor editing of English language required

Author Response

We are very grateful to the reviewers for their detailed and insightful
comments, and have revised the paper accordingly. Here are our replies:

2.1: "Most of the equations used in this work are known. They should be
properly cited. Any change in any equation or a newly proposed
equation should be clearly mentioned as "our proposed work" so that
the readers can differentiate between the contributions of the work
and the existing work."

We have emphasized the novelty in the paper, for both the relaxed
problem (Section 3) and the non-relaxed one (Section 4).

2.2: "Literature review is very weak and does not cover the domain in
depth and breadth. PNP is an extensively studied problem and one can
find hundreds of related works. However, the authors haven't covered
the literature in a meaningful manner."

We have added a survey and references (please see reply 1.1 to Reviewer 1).

2.3: "The proposed approach heavily relies on the existing work [2]
(section 2). I am not able to find the novelty of the proposed
work. What are the main contributions? Section 3.1.1 discusses the
proposed improvement, however, the novelty of the work is still
questionable."

We have done our best to better emphasize our novel contributions, both
in Section 3 (relaxed problem), for which we present a novel algorithm
which is much faster and requires less memory than previous work, and
Section 4 (non-relaxed case) in which we present a novel approach for
a fast solution with controllable, guaranteed accuracy. We chose [2]
as a reference, because (a) it rigorously tested many existing
methods, and (b) instead of using off-the-shelf minimizers, put a
great deal of effort into tailoring a fast solution specifically for
the PnP problem. Our running times are faster than those reported
in other recent papers, for example SQPnP ([10] in the revised version),
as well as in [9].

2.4: "The proposed approach is not tested on publicily available
datasets. The proposed approach heavily relies on [2] and the authors
also choose the same paper as benchmark. They also use the dataset
provided by [2]. The data, which is 230MBs, is not sufficient and the
findings on such a small dataset can be used to claim that the
proposed work is "10x faster and reliable". The proposed work is 10x
faster than [2] and only on the dataset provided by [2]."

We'd like to offer two replies to this remark. First, we believe
that a dataset of 230MB is quite large, and it is also rather
diverse. Second -- perhaps even more important, in our opinion --
is that the novel algorithms we offer are generic, and not data
dependent. Whatever the data is, eventually it yields a polynomial
optimization problem, and it is this problem we solve (in two
different approaches).

2.5: "Authors must test their proposed work on larger, extensive and
heterogenous publicly available datasets and also compare work with
other related works (not with [2] only)."

We believe that, as explained in 2.3 and 2.4 above, (a) the data
suffices, and (b) since [2] compared their approach with many
different methods, and we compare with [2], this demand is
already met. Further, please see remark in 2.4 re the solution
being generic.

2.6: "One expects other evaluation parameters as well e.g. computation
time, time complexity of the algorithm, space complexity etc. Authors
should also include specs of the system on which the algorithm was
tested."

We have provided detailed statistics of the running time, as well as
the space complexity (size of Hessian and linear systems which
should be solved in the SDP problem). We have also included
information on the platform used for implementation (i7-6500u processor).

2.7: "Section 4 should be merged with section 3 and the name should be
proposed work. Similarly all of the results should be in the results
section. Currently, the paper is in a very odd format."

We have added a new Section 5 with the results, but also figured that
putting two very different algorithms in the same section may be a bit
confusing, so we have left them in separate sections.

2.8: "Section 4.2: What dataset was used? Have you compared your findings
with any of the existing works? The claim 'guaranteed' is not
supported anywhere in the manuscript."

In Section 4 (Algorithm II), we have used the same dataset as in Section 3
(Algorithm I). A direct comparison to other work cannot be obtained, since
Algorithm II offers a novel approach to the non-relaxed problem, which,
to the best of our knowledge, did not exist before. We apologize if
'guaranteed' was mis-understood; we are referring to the approximation
guarantee, which is proved in Section 4.1 (revised version).

Author Response

We are very grateful for the detailed and helpful comments by the reviewer; we have carefully followed all the 11 requested corrections.

Reviewer 4 Report

The comments are in the uploaded file.

Author Response

We are very grateful to the reviewer, for the detailed and correct comments. We have carefully followed all of them. We note that for comment 1, we added a reference [15], which we believe is more suitable. 

Round 2

Reviewer 1 Report

All my previous concerns have been sufficiently addressed in the revision. In my opinion, the revised article is now suitable for publication.

Author Response

We are very grateful to the reviewer, both for the initial comments, which were very helpful in improving the paper, and for his/her final evaluation.